# Complex Brines and Their Implications for Habitability

**DOI:** 10.3390/life11080847

**Published:** 2021-08-19

**Authors:** Nilton O. Renno, Erik Fischer, Germán Martínez, Jennifer Hanley

**Affiliations:** 1Department of Climate and Space Sciences and Engineering, University of Michigan, Ann Arbor, MI 48109, USA; erikfis@umich.edu; 2Lunar and Planetary Institute, Universities Space Research Association, Houston, TX 75835, USA; gmartinez@lpi.usra.edu; 3Lowell Observatory, Flagstaff, AZ 86001, USA; jhanley@lowell.edu

**Keywords:** brines, complex brines, liquid water, habitability, habitable, life, icy worlds, Mars

## Abstract

There is evidence that life on Earth originated in cold saline waters around scorching hydrothermal vents, and that similar conditions might exist or have existed on Mars, Europa, Ganymede, Enceladus, and other worlds. Could potentially habitable complex brines with extremely low freezing temperatures exist in the shallow subsurface of these frigid worlds? Earth, Mars, and carbonaceous chondrites have similar bulk elemental abundances, but while the Earth is depleted in the most volatile elements, the Icy Worlds of the outer solar system are expected to be rich in them. The cooling of ionic solutions containing substances that likely exist in the Icy Worlds could form complex brines with the lowest eutectic temperature possible for the compounds available in them. Indeed, here, we show observational and theoretical evidence that even elements present in trace amounts in nature are concentrated by freeze–thaw cycles, and therefore contribute significantly to the formation of brine reservoirs that remain liquid throughout the year in some of the coldest places on Earth. This is interesting because the eutectic temperature of water–ammonia solutions can be as low as ~160 K, and significant fractions of the mass of the Icy Worlds are estimated to be water substance and ammonia. Thus, briny solutions with eutectic temperature of at least ~160 K could have formed where, historically, temperature have oscillated above and below ~160 K. We conclude that complex brines must exist in the shallow subsurface of Mars and the Icy Worlds, and that liquid saline water should be present where ice has existed, the temperature is above ~160 K, and evaporation and sublimation have been inhibited.

## 1. Introduction

One of the top goals of NASA’s space science program is to use knowledge of the history of the Earth, and the life on it, as a guide for determining the processes and conditions that create and maintain habitable environments beyond Earth [1]. Habitability or astrobiological potential are defined as the potential for an environment to support life, on scales ranging from microscopic to planetary-wide. In the past, to a large extent, assessments of astrobiological potential have focused on determining whether liquid water has been present on other worlds [1,2,3,4,5,6,7,8]. However, more specifically, the habitability of an environment depends on the presence of three main ingredients: (i) a solvent capable of supporting complex biochemistry [2,3], (ii) a source of energy to maintain the complex molecules, structures, and pathways on which life depends on [4], and (iii) nutrients and raw materials for biosynthesis [5]. Moreover, these ingredients must be available within environmental conditions amenable to the assembly, stability, and interaction of complex structures and molecules [1,6,7,8]. This article consists of a brief review of the literature on brines, the interpretation of observations, and predictions of the formation of complex brines based on thermodynamics.

As illustrated in Figure 1, brine inclusions produce habitable channels in terrestrial sea ice [9]. Similar features might have existed or might exist on Mars and in the Icy Worlds because brine inclusions always form when saline water freezes. As the temperature of a saline solution decreases, ice precipitates, increasing the salt concentration of the remaining liquid [10]. This process drives the concentration of saline solutions towards the value with the lowest freezing temperature possible, i.e., their eutectic values [11].

The eutectic temperature (*T_Eut_*) of binary aqueous solutions of salts already found on Mars is as low as ~199 K [11,12,13,14,15,16]. Eutectic solutions have salt mass fractions *χ_Eut_*~0.3–0.5 for salts such as NaCl and Ca(ClO_4_)_2_ [17]. The freezing point depression of saline solutions with respect to the value for pure water, Δ*T_Eut_*, depends on their composition, but even ubiquitous natural terrestrial solutions, such as seawater, remain partially liquid at temperatures as low as 230 K (Δ*T_Eut_* ≈ 43 K) [18]. Moreover, these solutions can remain liquid (metastable) at lower temperatures, while complex brines can have significantly lower eutectic temperatures than binary brines [19].

Earth, Mars, and carbonaceous chondrites have similar elemental abundances [20,21], but the Earth is depleted in the most volatile elements such as ammonia. The Icy Worlds, on the other hand, are expected to be richer in more volatile elements than the Earth because they formed further away from the sun [22,23,24]. Indeed, ammonia has been observed in Enceladus’ water plumes [25]. As our knowledge of the salts present on Mars and the Icy Worlds is very limited [14,22,24,25,26,27,28,29,30,31,32,33,34,35,36,37,38,39,40,41,42], we hypothesize that they have salts derived from elements found in carbonaceous chondrites [20], at least in trace amounts, and use this hypothesis together with our knowledge of the salts already discovered on Mars and the Icy Worlds [43,44,45] to study the complex brines that might exist on them.

The motivation for our approach to study complex brines on Mars and the Icy Worlds is that, as an ionic solution starts to freeze, ice and the least soluble compounds precipitate first, while the brine with the lowest possible eutectic temperature freezes last. This process drives the composition of aqueous solutions towards that of the brines with the lowest eutectic temperatures possible for the compounds available in the solutions [11]. Indeed, we postulate that this is the process that concentrates CaCl_2_ in Don Juan Pond in Antarctica (Figure 2) [46,47] and other reservoirs of liquid water found in Antarctica [47].

The fact that the eutectic temperature of ammonia–water solution is as low as ~160 K [24,48] is interesting because ~50% of the mass of the Icy Worlds is estimated to be water, and ~10% is estimated to be ammonia [22,40]. Moreover, the fact that the eutectic temperature of lithium–ammonia solution is ~90 K [49,50] could explain morphological features observed on the surface of Europa such as its chaotic terrains, and suggests that Europa could have aqueous solutions containing lithium and ammonia with eutectic temperatures between 90 and 160 K.

There is evidence that Recurring Slope Lineae (RSL), narrow low-reflectance features that form on present-day Mars [51,52], are likely caused by granular flow processes, but that hydration and perhaps deliquescence [53,54] might play a role in their formation. In fact, there is spectroscopic evidence that the amount of hydrated salts in RSLs increases when they become active [55]. This is interesting because it suggests that brines could be present in the shallow subsurface of Mars. Indeed, the freezing of a eutectic mixture most likely produced the soft ice found in the shallow subsurface of Mars by Phoenix [11,56,57,58,59].

Heinz et al. found that the yeast Debaryomyces hansenii is the most halotolerant microorganism discovered so far [60]. They argue that it is possible that potential Martian microbes could have adapted to higher salt concentrations than the terrestrial microorganisms we know because Martian microbes would have had a longer exposure to the briny environments occurring naturally on Mars but not on Earth. Indeed, if cold brines are ubiquitous on other worlds, life as we do not know would be expected to adapt to the coldest brines possible.

Here, we discuss the formation of the complex brines that could exist on Mars and the Icy Worlds. These brines are interesting because halophilic organisms thrive in terrestrial brines [61], even below 0 °C [62,63,64,65]; therefore, brines could form habitable environments on other worlds. Indeed, brine inclusions beneath a radiation-shielding ice layer could form habitable environments on the Icy Worlds. However, it is important to point out that metabolic processes in life as we know it have generally been detected only at temperatures as low as about −20 °C, and that life usually becomes dormant at lower temperatures, becoming active again only when temperatures rise above a higher threshold value. However, the viability of life as we do not know in the lowest temperature brines possible remains an open question.

## 2. Freeze–Thaw Cycles and the Formation of Complex Brines

Water vapor pressure at the triple point of water (~600 Pa) is below the present day atmospheric pressure at the lowest regions of Mars, such as the Phoenix, Curiosity, and Perseverance landing sites (~700–950 Pa), but the low surface temperature (~180–285 K), in combination with the extremely low water vapor partial pressure (not larger than ~1 Pa), inhibits the formation of pure liquid water [66,67,68,69,70,71] because it is thermodynamically unstable. However, liquid saline water (brine) can be present in the shallow subsurface where ground ice exists in contact with the saline regolith or in ice cracks and crevasses because even binary solutions of salts found on Mars depress the freezing point within the range of present-day temperatures of these regions [11,19,26,27,30,33,67,72,73,74].

Recurring freeze–thaw cycles can produce complex brines with the lowest eutectic temperature possible for the compounds available in a solution, potentially resulting in the formation of liquid brines in ice inclusions on Mars and the Icy Worlds. This could occur because the eutectic temperature of complex mixtures can be significantly lower than those of binary mixtures [19], which are already as low as ~199 K on Mars [13].

Figure 3 shows sketches of two- and three-dimensional versions of a ternary phase diagram illustrating the process that produces complex eutectic solutions. The crystallization of a mixture of composition X containing components A–C, such as, for example, NaCl, CaCl_2_ and H_2_O, leads to the formation of a ternary eutectic solution (at point E; Figure 3 (right). This eutectic solution is the portion of the mixture that freezes last when the temperature of the mixture is lowered slowly. Figure 3 (right) shows that, when the mixture of composition X intersects the liquidus surface, the crystals of C start to precipitate. If the temperature continues to be lowered, the composition of the liquid moves along a straight line away from C, while crystals of component C precipitate. Along this path, the liquid component becomes enriched in components A and B, while component C precipitates. When one of the boundaries (a binary eutectic curve) is reached at point O, crystals of A also start to precipitate, and the liquid follows the binary eutectic curve towards point M. Thus, the material that precipitates from O to M is a mixture of components A and C, in the proportion found at point P. Further cooling leads to additional precipitation until the formation of the ternary eutectic at point E.

Similar cooling processes can be used to experimentally determine the *n*-eutectic temperature of complex brines (*n*-components) because crystallization drives the concentration of a solution towards the *n*-eutectic value. The phase diagram does not need to be known in order for the range of environmental conditions in which complex brines could exist to be determined because the *n*-eutectic point gives the lowest temperature limit of this range. Once this eutectic point is found, the upper limit can be found by determining the partial pressure of the most volatile components of the mixture as a function of temperature. This can be conducted theoretically by knowing the composition of the *n*-eutectic. However, the most important implication of this discussion on phase diagrams is that complex brines with the lowest eutectic temperature possible for the chemicals available in an ionic solution form naturally via recurring freeze–thaw cycles, when the temperature of the solution cycles is around the lowest eutectic value possible for the chemicals available in the solution.

It follows from the discussion above those elements that exist in trace amounts on Earth, Mars, and the Icy Worlds such as lithium [75] can be concentrated by the precipitation of other compounds during the cooling of their aqueous solutions [10,11,45]. Indeed, lithium is most likely concentrated by precipitation of other compounds from brines present in the main terrestrial lithium reservoirs [76], and probably also from some of the clay minerals found on Mars [77]. Ice is expected to contain liquid brine inclusions in thermodynamic equilibrium with it, whenever the ice temperature is above the brine’s eutectic value [78]. This might have important implications for exobiology because some terrestrial microorganisms thrive in both ammonia-rich and lithium-rich waters [62,63].

## 3. Low Temperature Brine Candidates

Elements that exist in trace amounts on Mars and the Icy Worlds such as lithium can be concentrated by precipitation of other compounds by recurring freeze–thaw cycles [11]. This could lead to the formation of brines with the lowest possible eutectic temperature for the available chemical compounds. A list of salts relevant for formation of complex brines on Mars and the Icy Worlds is presented in Table 1. This list includes lithium and ammonia salts found in carbonaceous chondrites, but not identified on Mars or the Icy Worlds yet. The list in Table 1 is not exhaustive, it includes only the salts with either the lowest eutectic temperatures or largest concentrations expected on Mars or the Icy Worlds.

The data presented in Table 1 is interesting because it implies that the eutectic temperature of water–ammonia solutions can be as low as ~160 K, and a significant fraction of the mass of the Icy Worlds is estimated to be water substance and ammonia. Thus, brines with eutectic temperature of at least ~160 K are expected to have formed where, historically, the temperature has oscillated above and below ~160 K. Moreover, the fact that the eutectic temperature of lithium in ammonia is less than 100 K suggests that complex solutions of salts and ammonia in water with eutectic temperatures well below ~160 K may exist. Since the process that forms complex brines appears to be ubiquitous in nature [11,46,69], complex brines must exist in the shallow subsurface of Mars and the Ice Worlds; therefore, liquid saline water should be present where ice has existed, the temperature is the lowest eutectic temperature possible, and sublimation and evaporation have been inhibited.

## 4. Discussion

The fact that complex brines with eutectic temperatures below ~160 K may exist on Mars, the Icy Wolds, and beyond is interesting because a diverse array of terrestrial microorganisms thrives in brines, even in subglacial and deep seafloor habitats lacking sunlight [61,64,65]. The discovery of microbial communities in the subglacial brine that episodically drains from Taylor Glacier in Antarctica’s Dry Valleys [64] is particularly relevant to the understanding of the habitability of Mars. An overview of these briny habitats and the microorganisms that inhabit them has been presented in the literature [61]. Below, we summarize discoveries of microorganisms that could shed light on our understanding of the habitability of the Icy Worlds and Mars.

The discoveries of life in subsurface saline aquifers, deep-sea brine pools, and subglacial brine reservoirs [62,63,64,83,84,85] have expanded our knowledge of habitability. The cycling of sulfur, methane, and iron in briny habitats in the absence of sunlight underscores the metabolic flexibility of extremophiles [61]. The halophilic algae *Dunaliella Salina*, a member of the *Chlorophyceae* class of green algae and the first organism discovered in the Dead Sea [86,87,88], survives in brines with more than 20% salt content [61]. Halophilic archaea have been isolated from Dead Sea samples more than 50 years after their collection [89], indicating an extremely low mortality rate in these briny environments.

The extremely halophilic archaeon, *Haloquadratum Walsbyi*, discovered in a Red Sea salt pond [90], has been cultivated in a laboratory. This unusual microorganism floats in sun-lit, nutrient-rich brines. Halophilic organisms with salt-adapted enzymes are found in many branches of the tree of life, including archaea and bacteria. This suggests either that adaptation to saline environments occurred numerous times or that lateral gene transfer occurred during their evolution. Both mechanisms could explain the widespread distribution of carotenoid and rhodopsin pigments in halophilic microorganisms that gives red coloration to salt lakes, salterns, and salt flats.

As indicated in Table 2, brines form habitats for many microorganisms in cold regions. Microbes thrive in brine inclusions in sea ice, which concentrate nutrients and organics, making them readily available for microbial consumption [9]. Blood Falls, at the foot of the Taylor Glacier in Antarctica’s Dry Valleys, is a ‘brine-fall’ of iron(II)-rich fluid that contains interesting microorganisms such as the sulfur oxidizer *Thiomicrospira Arctica* [65]. Don Juan Pond and Taylor Glacier are good analogues for potentially habitable environments on Mars and the Icy Worlds.

On the sea floor, mud volcanoes, brine lakes, and anoxic basins exist in the Black Sea, the Red Sea, the Mediterranean Sea, and in the Gulf of Mexico. Life thrives in these seafloor brines that form when subsurface fluids encounter ancient, buried salt deposits [61].

## 5. Conclusions

Since the process that forms complex brines appears to be ubiquitous in nature [11,46,47], complex brines are expected to exist in the shallow subsurface of Mars, the Ice Worlds, and beyond. Indeed, briny water–ammonia solutions with eutectic temperatures below ~160 K are expected to have formed where, historically, subsurface temperatures of the Ice Worlds and Mars have oscillated above and below ~160 K. Furthermore, because the eutectic temperature of Li-NH_3_ is less than 100 K, complex brines with much lower eutectic temperatures than 160 K may exist. This might have important implications for exobiology because some terrestrial microorganisms thrive in both ammonia-rich and lithium-rich waters [62,63]. The fact that the eutectic temperature of lithium–ammonia solution is ~90 K [49,50] could explain morphological features observed on the surface of Europa, such as its chaotic terrains. Indeed, Europa could have aqueous solutions containing lithium and ammonia with eutectic temperatures between 90 and 160 K.

## Figures and Tables

**Figure 1 life-11-00847-f001:**
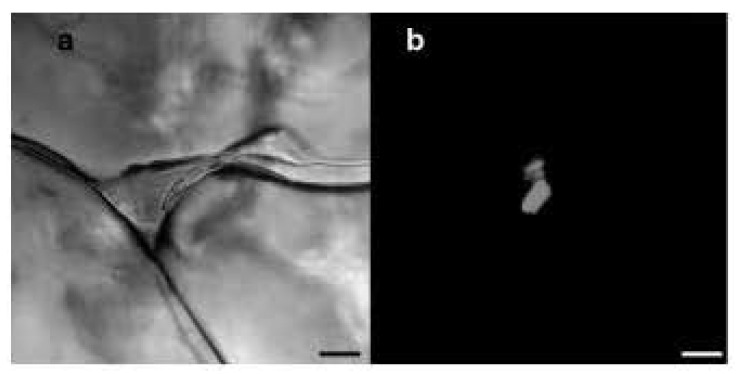
(**a**) Brine inclusions in sea ice containing two bacteria inside a triple junction between ice crystals. The scale bar at the bottom of the image is 10 μm long. (**b**) Epifluorescent image highlighting the presence of the two bacteria in the junction. After Junge et al., 2001 [9].

**Figure 2 life-11-00847-f002:**
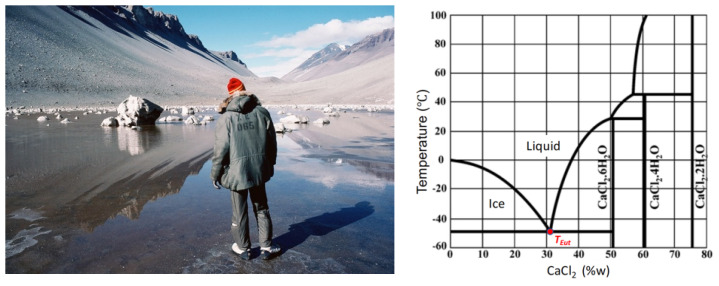
(**Left**) Don Juan Pond in Antarctica is a CaCl_2_-rich brine lake [46]. Credit: Ralph Lewis. (**Right**) Phase diagram indicating that the eutectic temperature of CaCl_2_ is −50 °C [47], Don Juan Pond’s typical winter temperature [46].

**Figure 3 life-11-00847-f003:**
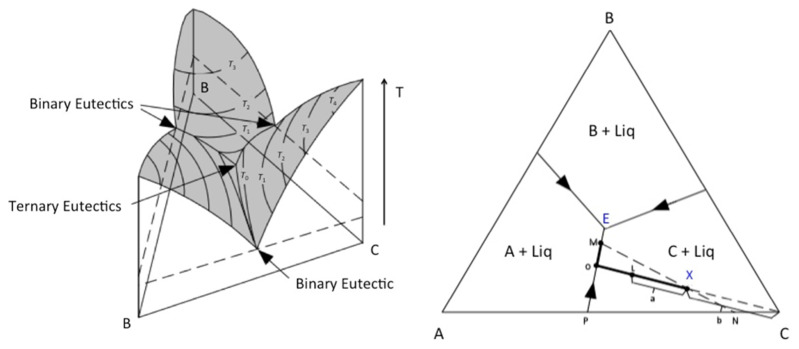
Three- and two-dimensional sketches of a ternary phase diagram: (**Left**) liquidus surfaces colored in gray, with composition plotted along the sides of the triangular base, and temperature plotted vertically. Isotherms are shown as contours on the gray liquidus surfaces. The binary eutectics are represented by the boundary curves between the surfaces that the two phases on each side of the curve crystalizes. (**Right**) Crystallization path of mixture of composition X, showing that the solids resulting from the crystallization of mixture X (bold path) contain components A–C.

**Table 1 life-11-00847-t001:** Salts found on Mars, the Icy Worlds, and in carbonaceous chondrites (LiBr and NH_3_) and their eutectic temperatures in water solutions. Lithium–ammonia solution (Li-NH_3_) is included in our list to highlight its extremely low eutectic temperature and the fact that it can exist in significant quantities in the Icy Worlds. The eutectic temperatures of the aqueous solutions of the various salts are from the literature ^a^ [8], ^b^ [79], ^c^ [80], ^d^ [81], ^e^ [23,82], and ^f^ [50].

Mars	*T_e_* in Water Solution (K)	Icy Worlds	*T_e_* in Water Solution (K)
Ca(ClO_4_)_2_	199 ^a^	NaCl	251 ^d^
Mg(ClO_4_)_2_	206 ^b^	MgCl_2_	240 ^d^
Mg(ClO_3_)_2_	204 ^c^	MgSO_4_	269 ^d^
Fe_2_(SO_4_)_3_	247 ^d^	KCl	262 ^e^
Ca(ClO_3_)_2_	232 ^c^	H_2_ SO_4_	186 ^f^
CaCl_2_	223 ^d^	Mg(ClO_4_)_2_	212 ^b^
MgSO_4_	269 ^d^	LiBr	201 ^d^
LiBr	201 ^d^	NH_3_	~160–170 ^e^
NH_3_	~160–170 ^e^	Li-NH_3_	~90 ^f^

**Table 2 life-11-00847-t002:** Some of the environments with the highest salt concentrations found on Earth, some of the microorganism found on them, and the results of record setting salt concentrations for life viability in laboratory experiments reported in the literature.

Environment	Salt and Salt Concentration	Microorganisms
Atacama’s salt crusts [91]	Nearly saturated NaCl	Halobacteriales, Bacteroidetes, Algae, and Cyanobacterial
Dead Sea [86]	Ca, Mg and Na (nearly saturated) Chlorides	Algae (*Dunaliella Salina*)
Spotted Lake [92]	Nearly saturated Sulfates	Archaea, uncharacterized Bacteria, and Cyanobacteria
Don Juan Pond [2]	Saturated CaCl_2_	Yeasts, Algae, Fungi and Bacteria
Discovery Basin [93]	Nearly saturated MgCl_2_	Protists and Fungi
L’Atalante Basin [93]	Na and K Sulfates	Protists and Fungi
Laboratory [60,94]	Perchlorates at 9%	Archea (*Halorubrum lacusprofundi*)
Laboratory [60,95]	NaClO4 at 12%	Bacteria (*Planococcus halocryophilus*)
Laboratory [60]	NaClO4 at 20%	Fungi (*Debaryomyces hansenii*)

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
