# Peer review of "Complex Brines and Their Implications for Habitability"

_life, 2021, doi:10.3390/life11080847_

Round 1

Reviewer 1 Report

This was an interesting paper, and highly readable for someone outside the field.

Author Response

We would like to thank the reviewers for the positive feedback.

Reviewer 2 Report

I enjoyed reading the short paper which is very informative. The one major concern I have is that the paper seems to imply that very cold brines are habitable and that this is only a function of life´s adaptabilities to high salt concentrations (and the solution to staying liquid). However, if life is possible in these cold brines, the (micro)organisms would also have to be adapted to cold temperatures, in essence they would have to be multi-extremophiles.

There is a reason why metabolic processes have generally only be detected at temperatures down to about -20 C (though there are some papers claiming activity at lower tempeatures). The main problem are energetic issues and life as we know it usually overcomes these by becoming dormant and then viable again after temperatures rise above a certain threshold temperature. I think this would have to be pointed out and emphasized in the discussion. In my view it would make the paper so much better (which is an excellent discussion of complex brines otherwise), especially in a journal that is called Life.

One other, more minor, suggestion: it would be helpful to the reader what the actual microbial record holders are in regard to adaptability to salt brines. A table may be useful here. There has also been a recent paper that lists those for sodium perchlorate, which seems relevant to the discussion given the eutectic: Heinz et al. (2020) A new record for microbial perchlorate tolerance: fungal growth in NaClO4 Brines and its implications for putative life on Mars. Life 10 (5): 53, https://doi.org/10.3390/life10050053. It would make sense to cite it in lines 103-107 and discuss it and/or other related papers on what microbial life as we know it can manage in cold brines (not only up to what temperature the solution can stay liquid) - not to say that life as we don´t know it may have a higher tolerance. I think it would be helpful if some of these thoughts and considerations are spelled out.

Author Response

We would like to thank the reviewer for the excellent suggestions.

We point out in the revised article (lines 114-119) that metabolic processes in life as we know have generally been detected only at temperatures as low as about -20 C and that life usually  becomes dormant at lower temperatures, becoming active again only when the temperature rise above a higher threshold value. However, the viability of life as we don’t know in brines at lower temperatures remains an open question.

We also point out in the revised article that (lines 103-109): Heinz et al. (2020) found that the yeast Debaryomyces hansenii is the most halotolerant microorganism discovered so far.  They argue that it is possible that potential Martian microbes could have adapted to higher salt concentrations than the terrestrial microorganisms we know because they would have had a long exposure to the briny environments occurring naturally on Mars but not on Earth. Indeed, if cold brines is ubiquitous in other worlds, life as we don’t know would be expected to adapt to the coldest brines possible.

We added the Table 2 in the discussion section summarizing the microbial record holders regarding adaptability to brines.

Reviewer 3 Report

The paper titled "Complex brines and their implications for habitability" addresses the possibility that the fractures due to the complex brines that can exist on Mars and icy moons like Europa, Ganymede, etc can accommodate life. The authors discuss how these complex brines form, what are the possible brine candidates, and the possible microorganism that can strive in those conditions. The Brief report addresses a timing topic proposing a possible habitat in these cold worlds. 
I recommend the brief report for publication. In reading, I just found the following:

- Figures are too small, especially the two panels of Figure 3. They are difficult to read. The several points addressed in the text discussion (since row 127) are not even visible.
- row 126 Figure 4 right --> Figure 3 right. No figure 4 exists.

that I recommend to the authors to mend. I do not need to review again the brief report.

Author Response

We would like to thank the reviewer for the corrections and suggestions. We increased the size of the figures.  We corrected the figure number from 4 to 3. 

Round 2

Reviewer 2 Report

All my concerns have been addressed. Thank you.